# Frequent detection of functional hyposplenism via assessment of pitted erythrocytes in patients with advanced liver cirrhosis

**Malte H. Wehmeyer**[1]* , **Harsha Sekhri**[1] , **Raluca Wroblewski**[2], **Antonio Galante**[3], **Thomas Meyer**[4], **Ansgar W. Lohse**[1], **Julian Schulze zur Wiesch**[1]

**1** Department of Medicine, University Medical Center Hamburg-Eppendorf, Hamburg, Germany, **2** Institute for Hematopathology Hamburg, Hamburg, Germany, **3** Gastroenterology and Hepatology, Ente Ospedaliero Cantonale, Università della Svizzera Italiana, Lugano, Switzerland, **4** Department of Dermatology, Venerology and Allergology, Ruhr-University Bochum, Bochum, Germany

☯ These authors contributed equally to this work.
* m.wehmeyer@uke.de

**Data Availability Statement:** Data (patient level) cannot be shared publicly because of legal reasons

## Abstract

### Background

Asplenia or functional hyposplenism are risk factors for severe infections, and vaccinations against encapsulated bacteria are advised. There are only limited data regarding the spleen function of cirrhotic patients.

### Methods

We evaluated spleen function in patients with liver cirrhosis, who were prospectively enrolled in this study. Spleen function was evaluated by the measurement of pitted erythrocytes. Functional hyposplenism was defined as a percentage of PE of >15%.

### Results

117 patients, mean age 58.4 years and 61.5% (n = 72) male with liver cirrhosis were included. Functional hyposplenism was diagnosed in 28/117 patients (23.9%). Pitted erythrocytes correlated with albumin (p = 0.024), bilirubin (p<0.001), international normalized ratio (INR; p = 0.004), model of end-stage liver disease (MELD) score (p<0.001) and liver stiffness (p = 0.011). Patients with functional hyposplenism had higher MELD scores (median 13 vs. 10; p = 0.021), liver stiffness (46.4 kPa vs. 26.3 kPa; p = 0.011), INR (1.3 vs. 1.2; p = 0.008) and a higher Child-Pugh stage (Child C in 32.1% vs. 11.2%; p = 0.019) as compared to patients without functional hyposplenism. Functional hyposplenism was not associated with the etiology of cirrhosis. Importantly, 9/19 patients with Child C cirrhosis had functional hyposplenism.

(data protection acts in Germany). However, anonymized data are available from the authors (Corresponding author) on reasonable request. Requests for anonymized data may also be sent to direktion1med@uke.de (office of the I. Department of Medicine, University Medical Center Hamburg-Eppendorf).

**Funding:** The author(s) received no specific funding for this work.

**Competing interests:** The authors have declared that no competing interests exist.

## Conclusion

A quarter of patients with liver cirrhosis and almost 50% of patients with Child C cirrhosis have functional hyposplenism. Functional hyposplenism is associated with poor liver function and the degree of portal hypertension, which is characterized by higher liver stiffness measurements in transient elastography.

## Introduction

More than one million deaths are caused by liver cirrhosis worldwide per year [1]. Patients with liver cirrhosis are considered at increased risk for infectious complications, such as spontaneous bacterial peritonitis (SBP) [2]. This vulnerability to infectious complications has been mainly accredited to the cirrhosis-associated immune dysfunction syndrome (CAIDS) [3, 4] due to decreased reticuloendothelial cells, phagocytic activity, and neutrophil mobilization, as well as increased bacterial translocation from the gut [3, 5]. Functional hyposplenism (FH), which is a risk factor for severe infections caused by encapsulated bacteria [6], has also been observed in patients with alcoholic liver disease [7–9]. FH is also prevalent in other diseases, such as celiac disease, inflammatory bowel disease, autoimmune disease, amyloidosis and sickle cell disease [10–15]. Official vaccinations recommendations for patients with FH include influenza, pneumococcal, meningococcus and Haemophilus influenza [16]. However, current vaccination guidelines for patients with liver cirrhosis do not cover pneumococcal or meningococcus vaccinations [16]. There are only limited data on the prevalence of FH in liver cirrhosis and its predisposing factors. The frequency of pitted erythrocytes in the peripheral blood smear is considered the gold standard for the evaluation of the splenic function [6, 14, 17, 18]. Pits represent vacuoles attached to the cell membrane containing debris like ferritin, hemoglobin, membranes, and remains of mitochondria which would have been normally removed by an intact spleen [19]. Previously, the frequency of pitted erythrocytes was shown to correlate inversely with the functional splenic volume in patients with sickle cell disease, splenectomized patients, and healthy controls [18]. In this study, functional splenic volume was measured by $^{99m}$Tc-labeled autologous erythrocyte scintigraphy, clearance of labeled erythrocytes and low-dose computed tomography (to evaluate the total splenic volume). Therefore, pitted erythrocytes are regarded as a reliable indirect marker of functional hyposplenism [6, 13–15]. Clinically relevant FH is expected at a threshold of 15% pitted erythrocytes [6, 17, 18, 20], while a value of a maximum of 4% pitted erythrocytes is considered physiologic [6, 17, 20].

The occurrence of Howell Jolly bodies (HJB) in peripheral blood smears, which represent DNA remnants in erythrocytes from progenitor cells is also associated with FH [6, 13, 14, 17, 21]. However, HJB are less sensitive for the evaluation of a mild splenic dysfunction [18, 22–24].

The aim of our study is to determine the occurrence of FH in patients with liver cirrhosis and to identify possible risk factors for impaired splenic function in these patients.

## Patients and methods

The University Medical Center Hamburg-Eppendorf is a large medical center in northern Germany that covers a catchment area of more than 5 million inhabitants. As a tertiary referral center, it provides full service in all fields of medicine, including liver transplantation. At our outpatient clinic for hepatology, we annually see approximately 500 patients with liver cirrhosis.

For this current study, 117 patients with liver cirrhosis were prospectively enrolled from June 2016 to June 2017. Patients with hematological co-morbidities or with a history of splenectomy were not included in the cohort of patients with liver cirrhosis. The study was approved by the local ethics committee (Ethikkommission Ärztekammer Hamburg, reference number PV4081) in accordance with the principles of the declaration of Helsinki and all patients gave written informed consent. For all patients, data on demographics, etiology of liver cirrhosis, hepatocellular carcinoma (HCC), and other complications of cirrhosis, co-morbidities, vaccination status, and laboratory data were collected. All patients at our center receive a standardized etiologic work-up during the first admission at our outpatients unit: anamnesis and ethyl glucoronide in urine to evaluate alcohol misuse, hepatitis B- and C-serology, evaluation of liver specific auto-antibodies, as well as measurements of coeruloplasmin levels, transferrin saturation and alfa-1-antitrypsin levels. If necessary, a transcutaneous or mini-laparoscopically guided liver biopsy is performed. Furthermore, data from transient elastography (TE) and ultrasound were collected. Follow-up data of the patients (infections during follow-up, last contact, as well as date and cause of death) were collected in April 2022 by chart review.

## Assessment of pitted erythrocytes and Howell-Jolly bodies

The splenic function was assessed by inspection of a peripheral blood smear. The blood smear was prepared on the same day the blood sample was obtained. Erythrocytes with an abnormal membrane, such as indentations or pits ("pitted erythrocytes") were counted by employing the method described by Corazza [23] and Muller [7]. A drop of venous blood and 0.5 ml of 3% glutaraldehyde were mixed in an Eppendorf tube. Then 0.4 ml of the solution was examined using interference contrast microscopy (microscope Primovert, Zeiss, Jena, Germany). With 60x oil immersion optics, 10 fields of view were photographed (Primovert HDcam, Zeiss, Jena, Germany) and assessed for each patient by one blinded investigator. The visible erythrocytes and pitted erythrocytes were counted (Fig 1). The percentage of erythrocytes with these abnormalities was calculated (the number of pitted erythrocytes divided by the total number of erythrocytes). A threshold of 4% was defined as the upper limit of normal, patients with pitted erythrocytes percentage > 15% were defined as suffering from functional hyposplenism (FH). Using the same peripheral blood smear Howell-Jolly bodies were counted. For comparison

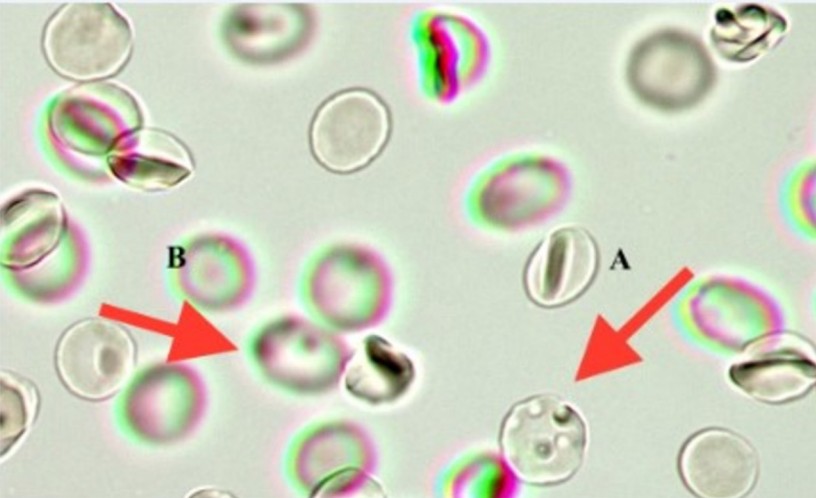

**Fig 1.** Image of pitted erythrocytes (indicated by arrow A) and Howell-Jolly bodies (indicated by arrow B).

and validation, pitted erythrocytes were determined in 5 splenectomized patients and a control group of 15 healthy subjects.

## Statistics

Categorical variables were analyzed using Fisher's exact test and continuous variables were analyzed by Mann-Whitney-U test (for non-normally distributed variables) or t-test (for normally distributed variables, e.g. age). Pearson Bravais test was used for correlation analysis. To identify independent predictors of FH, variables with a p-value of <0.1 in the univariate analysis were entered into a stepwise backward logistic regression model. A Cox regression model was used to analyze the follow-up data. Patients were censored if they received a liver transplantation. The primary endpoint for the Cox regression analysis was infection or death (whichever occurred first). Secondary endpoints included infection-related mortality and all-cause mortality. Statistical analyses were conducted using SPSS Statistics Version 22, graphs were created using Graph Pad Prism 4.

## Results

### Patients' characteristics

In total, 117 patients were included in the study. The mean age was 58.4 (standard deviation ± 11.3 years) and 61.5% (n = 72) were male. Child A cirrhosis was diagnosed in 63 patients (53.8%), 35 patients had Child B cirrhosis (29.9%) and 19 patients had Child C cirrhosis (16.2%). The median model for end-stage liver disease (MELD) score was 11 (range 6 to 27). The causes of liver cirrhosis or fibrosis were (multiple etiologies for each patient were possible): alcohol (58.1%; n = 68), hepatitis B virus (HBV) or hepatitis C virus (HCV) infection (20.5%; n = 24), non-alcoholic steatohepatitis (NASH; 14.5%; n = 17) and others (12.0%; n = 14; including 5 patients with cryptogenic cirrhosis, 4 patients with autoimmune hepatitis, 1 patient with hemochromatosis, 1 patient with alfa 1-antitrypsin deficiency, 1 patient with secondary sclerosing cholangitis, 1 patient with cirrhose cardiaque and 1 patient with Wilson's disease). Of these, five patients had alcoholic cirrhosis with concomitant chronic HBV or HCV infection and one patient had alcoholic cirrhosis and Wilson's disease. Sixteen patients (13.7%) had an HCC or a history of HCC, 68 patients had esophageal or gastric varices (58.1%), 37 patients had ascites (31.6%) and 28 patients had previously undergone a TIPS procedure (transjugular intrahepatic portosystemic shunt; 23.9%). In total, only 69 patients (59%) had a vaccination card and only 24/69 patients (34.8%) had a documented vaccination against pneumococci.

### Spleen function

In total, in 28/117 patients (23.9%) FH was detectable. 14 of 15 healthy controls had a pitted erythrocyte count < 4% (one subject had 4.4%). Four of five splenectomized patients had pitted counts of ≥ 15%, and one patient had 3.3%. The median pitted erythrocyte count in the cirrhotic patients was 10.8% (3.6%– 57.0%) compared to 2.1% (0.7% - 4.4%) in the healthy controls (p<0.001) and up to 15.7% (3.3%– 25.8%) in the group of splenectomized patients (p = 0.1835). Patients with a history of splenectomy had significantly higher frequencies of circulating pitted erythrocytes as compared to healthy controls (p = 0.002; **Fig 2A**), too. Importantly, only 2/117 (1.7%) of the cirrhotic patients had pitted erythrocyte counts below the upper limit of normal (4%). One patient had alcoholic cirrhosis, while the other patient had NASH-associated liver cirrhosis. Both patients had Child A cirrhosis.

In total, FH was diagnosed in 19/68 patients with alcoholic cirrhosis (27.9%), 4/24 patients with viral hepatitis (16.7%; including one patient with concomitant alcoholic misuse), 1/17

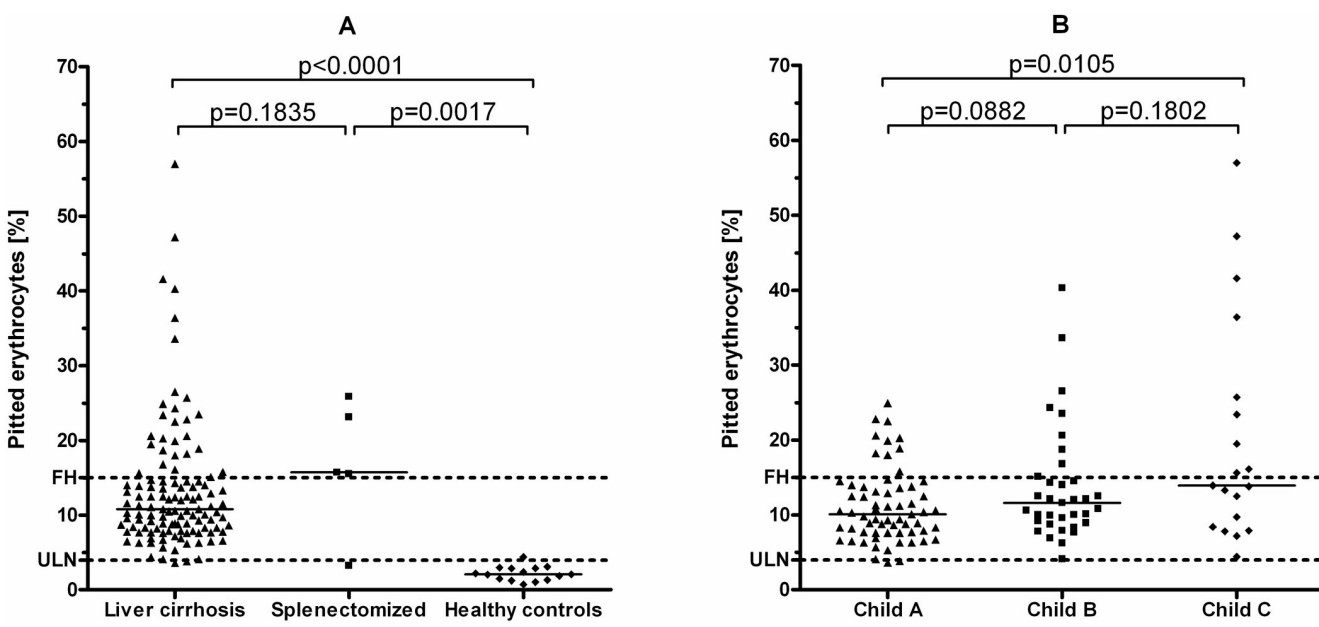

**Fig 2.** Comparison of pitted erythrocytes in cirrhotic patients, splenectomized patients and healthy controls (A) and between patients with Child A, B and C cirrhosis, respectively (B). The dotted lines indicate the upper limit of normal (4% pitted erythrocytes) and the threshold for diagnosis of functional hyposplenism (15% pitted erythrocytes). Results from the Mann-Whitney-U test are also shown. [ULN = upper limit of normal; FH = functional hyposplenism].

patients with NASH (5.9%), and 5/14 patients (35.7%) with another etiology of cirrhosis (3/4 patients with autoimmune hepatitis, 1/1 patient with alfa 1-antitrypsin deficiency and 1/1 patient with cirrhose cardiaque).

Overall, 10/63 (15.9%) patients with Child A cirrhosis, 9/35 (25.7%) patients with Child B cirrhosis, and 9/19 (47.4%) with Child C cirrhosis had FH (p = 0.019). Patients with Child C cirrhosis had significantly higher pitted erythrocyte counts as compared to patients with Child A cirrhosis (p = 0.011; **Fig 2B**). More patients with current ascites (32.4%; n = 12/37) as compared to patients without ascites at the time of study inclusion (20%; n = 16/80) had FH, but the association was not statistically significant (p = 0.166). Also, the presence of splenomegaly (defined as a spleen length > 120mm, as measured by ultrasound) was not associated with the presence of FH (p = 0.628).

The percentage of pitted erythrocytes in the peripheral blood correlated with albumin levels (r = -0.208; p = 0.024), bilirubin levels (r = 0.319; p<0.001), the international normalized ratio (INR; r = 0.268; p = 0.004), the MELD score (r = 0.339; p<0.001) and with the percentage of HJB in the peripheral blood smear (r = 0.770; P<0.001). We found no statistically significant correlation between pitted erythrocyte count and serum creatinine (r = 0.081; p = 0.385), aspartate aminotransferase (ASAT; r = 0.058; p = 0.533) or alanine aminotransferase (ALAT; r = -0.059; p = 0.530) levels, respectively. Furthermore, the percentage of pitted erythrocytes did not correlate with the spleen length as measured by ultrasound (r = -0.031; p = 0.736).

MELD score (13 [6–27] vs. 10 [6–29]; p = 0.021), INR (1.3 [1.0–3.0] vs. 1.2 [0.9–2.7]; p = 0.008) and HJB levels (11.7% [1.5%-59.3%] vs. 5.7% [0.1%-46.7%]; p = 0.011) were significantly higher in patients with FH as compared to patients without FH. Median bilirubin levels were higher in patients with FH as compared to patients without FH, but the results were statistically not significant (1.7mg/dl [0.4–17.0] vs. 1.0mg/dl [0.3–13.2]; p = 0.070). Further details are found in **Table 1**.

## Association of portal hypertension with functional hyposplenism

To study the impact of portal hypertension on spleen function, we compared the presence of ascites and/or esophageal (or gastric) varices at the time of study inclusion between patients with FH and patients without FH. However, neither ascites (p = 0.166) nor the presence of varices (p = 0.506) was associated with FH in our cohort (**Table 1**). Also, the percentage of PE did not correlate with the platelet counts (r = 0.101; p = 0.279) and there was no significant association of platelet counts with the presence of FH (p = 0.144). As liver stiffness measured by transient elastography (TE) is considered a marker for portal hypertension, we compared TE measurements between patients with FH to patients without FH: TE measurements in patients with FH were significantly higher as compared to patients without FH (46.4 kPa [13.5–75] vs. 26.3 kPa [3.8–75]; p = 0.011) and TE measurements showed a positive correlation with PE counts (r = 0.370; p = 0.005).

**Table 1. Comparison of patients with and without functional hyposplenism.**

| Parameter | | No hyposplenism (n = 89) Mean ± SD; median (range); N (%) | Functional hyposplenism (n = 28) Mean ± SD; median (range); N (%) | *P* |
|---|---|---|---|---|
| Age [years] | | 58.46 ± 11.6 | 58.0 ± 10.6 | 0.845 |
| Male sex | | 58 (65.2%) | 14 (50%) | 0.183 |
| Child-Pugh stadium | | | | 0.019* |
| | *Child A* | 53 (59.6%) | 10 (35.7%) | |
| | *Child B* | 26 (29.2%) | 9 (32.1%) | |
| | *Child C* | 10 (11.2%) | 9 (32.1%) | |
| Diabetes mellitus | | 26 (29.2%) | 6 (21.4%) | 0.476 |
| Etiology | | | | |
| | *Alcohol* | 49 (55.1%) | 19 (67.9%) | 0.276 |
| | *Viral* | 20 (22.5%) | 4 (14.3%) | 0.430 |
| | *NASH* | 16 (18.0%) | 1 (3.6%) | 0.069 |
| | *Other* | 9 (10.1%) | 5 (17.9%) | 0.318 |
| TIPS | | 21 (23.6%) | 7 (25%) | 1 |
| Ascites | | 25 (28.1%) | 12 (42.9%) | 0.166 |
| Varices | | 52 (58.4%) | 19 (67.9%) | 0.506 |
| Portal vein thrombosis | | 12 (13.5%) | 4 (14.3%) | 1 |
| HCC | | 10 (11.2%) | 6 (21.4%) | 0.208 |
| Splenomegaly | | 65 (73.0%) | 22 (78.6%) | 0.628 |
| Spleen length [mm] | | 134 (81–210) | 132 (88–193) | 0.893 |
| Transient elastography [kPa] (n = 55) | | 26.3 (3.8–75.0) | 46.4 (13.5–75.0) | 0.011* |
| Platelets [x10$^9$/l] | | 130 (23–632) | 99 (30–384) | 0.144 |
| ASAT [U/l] | | 40 (13–251) | 50 (19–207) | 0.259 |
| ALAT [U/l] | | 33 (12–187) | 33 (14–197) | 0.976 |
| Albumin [g/l] | | 33 (17–45) | 30 (16–48) | 0.139 |
| Bilirubin [mg/dl] | | 1.0 (0.3–13.2) | 1.7 (0.4–17.0) | 0.070 |
| Creatinine [mg/dl] | | 0.89 (0.46–2.90) | 0.88 (0.55–4.30) | 0.818 |
| INR | | 1.2 (0.9–2.7) | 1.3 (1.0–3.3) | 0.008* |
| MELD score | | 10 (6–27) | 13 (6–27) | 0.021* |
| Howell-Jolly Bodies [%] | | 5.7 (0.1–46.7) | 11.7 (1.5–59.3) | 0.011* |

n = number; SD = standard deviation; NASH = non-alcoholic steatohepatitis; TIPS = transjugular intrahepatic portosystemic shunt; HCC = hepatocellular carcinoma; ASAT = aspartate aminotransferase; ALAT = alanine aminotransferase; INR = international normalized ratio; MELD = model for end-stage liver disease;
* p < 0.05.

According to the Baveno VII consensus guidelines [25], cirrhotic patients with portal hypertension can be identified by the following criteria:

i.  TE measurement $\geq$ 25 kPa, or

ii.  TE measurement $\geq$ 20 kPa and platelet count $<$ 150 billion/l, or

iii.  TE measurement $\geq$ 15 kPa and platelet count $<$ 110 billion/l

In total, 38/55 patients (69.1%) of the patients were identified as suffering from portal hypertension according to these criteria. PE counts were significantly higher in patients with portal hypertension (12.7%, 5.3–57.0%) as compared to patients without portal hypertension (8.9%, 3.6–15.8%; p = 0.008).

In a subset of patients (n = 13) the hepatic venous pressure gradient (HVPG) was available from the time of study inclusion (or up to 3 months before study inclusion): The median HVPG of patients with FH (n = 5) was 30 mmHg (20–38), as compared to a median HVPG of 22 mmHg (17–32) in patients without FH (n = 8; p = 0.180).

### Multivariate analysis of possible predictors of functional hyposplenism

In the multivariate logistic regression model, only the MELD score was an independent predictor of FH in patients with liver cirrhosis (odds ratio [OR] 1.21; 95% confidence interval [95%CI] 1.06–1.39; p = 0.006).

### Follow-up data and vaccinations rates

Follow-up data were available from 112 patients. The median follow-up time was 2.57 years (range 0.02–5.78 years). 28/112 patients (25%) died during the follow-up period (all-cause mortality), 10/112 patients (8.9%) died due to infection-related complications (infection-related mortality), and 14/112 patients (12.5%) received a liver transplantation and were censored at this timepoint. In total, 9/28 patients with FH (32.1%) compared to 26/84 patients without FH (31.0%; p = 1) developed any bacterial infection during the follow-up period. In the Cox regression analysis, FH was not associated with the occurrence of the primary endpoint (infection or death; OR 0.64; 95%CI 0.61–2.25; p = 0.638), infection-related mortality (OR 0.91; 95%CI 0.19–4.23; p = 0.902), or all-cause mortality (OR 0.77; 95%CI 0.29–2.03; p = 0.594) in our study population.

In a multivariate (backward stepwise) Cox regression model including FH and MELD, only the MELD score was associated with occurrence of the primary endpoint (OR 1.18; 95%CI 1.11–1.25; p<0.001), infection-related mortality (OR 1.23; 95%CU 1.09–1.40; p = 0.001), and all-cause mortality (OR 1.22; 95%CI 1.13–1.31; p<0.001).

There was no significant difference with respect to the vaccination rates against pneumococci between patients with FH and without FH (5/17 [29.4%] vs. 19/52 [36.5%]; p = 0.771).

## Discussion

To our knowledge, this study is the largest study investigating the frequency and predictors of functional hyposplenism in patients with liver cirrhosis. Our investigation of the splenic function of cirrhotic patients by measurement of pitted erythrocytes as a possible indicator for compromised immune function showed that a large number of patients with cirrhosis of different etiologies have signs of functional hyposplenism and would formally need to be vaccinated against encapsulated bacteria. However, only a minority of our patients had a documented vaccination against pneumococci.

Only 2/117 patients with cirrhosis had pitted erythrocyte counts below the upper limit of normal (< 4%). A pitted erythrocyte count of > 4% may also indicate an impaired immune response against encapsulated bacteria. However, asplenic patients or patients with clinically relevant hyposplenism (e.g. due to sickle cell disease) were reported to have pitted erythrocyte counts >15% [6, 17, 20]. Notably, every second patient with Child C cirrhosis in our cohort had pitted erythrocyte counts > 15%. Therefore, the grade of functional hyposplenism in patients with advanced liver cirrhosis seems comparable to splenectomized patients.

In previous studies [7–9] functional hyposplenism was mostly observed in patients with alcoholic liver disease. While elevated pitted erythrocyte counts above the upper limit of normal were reported in liver healthy individuals with chronic alcohol misuse, functional hyposplenism (as defined by pitted erythrocytes > 15%) was not observed in these patients [26]. However, we found that also patients with liver cirrhosis due to viral hepatitis, NASH, autoimmune hepatitis, alfa1-antitrypsin-deficiency, and cirrhose cardiaque may suffer from functional hyposplenism. Most recently, pitted erythrocyte counts above the upper limit of normal (> 4%) were observed in two-thirds of patients with autoimmune liver diseases (66.6%; autoimmune hepatitis, primary sclerosing cholangitis and primary biliary cholangitis) [27], but the authors did not describe the cirrhosis status of these patients. Furthermore, only a minority of the patients in this study had functional hyposplenism (as defined by pitted erythrocytes > 15%) [27]. Interestingly, resolution of functional hyposplenism was reported in a small cohort of cirrhotic patients after liver transplantation [28]. Therefore, we hypothesize that functional hyposplenism is not only related to the underlying liver disease but the presence of cirrhosis and portal hypertension itself.

We found a positive correlation between the percentage of pitted erythrocytes in the peripheral blood smear and the INR, bilirubin levels, and the MELD score, as well as a negative correlation with albumin levels. Furthermore, patients with functional hyposplenism had more advanced cirrhosis considering MELD scores and the Child-Pugh stadium of the patients. These findings indicate a possible relation between poor liver function and functional hyposplenism. A possible association between liver and spleen function has been reported before, at least in patients with alcoholic liver cirrhosis [9]. Moreover, the correlation of liver stiffness with the percentage of pitted erythrocytes and higher liver stiffness in patients with functional hyposplenism may indicate a possible association between portal hypertension and spleen function. Patients with suspected portal hypertension according to non-invasive criteria [25] had a significantly higher percentage of pitted erythrocytes as compared to patients without portal hypertension. However, previous studies found that functional hyposplenism in patients with non-cirrhotic portal hypertension is uncommon [9] and we found no significant association between spleen function and spleen size, platelet count, the presence of ascites, or the presence of esophageal (or gastric) varices, respectively. The non-correlation of spleen size (or spleen volume) and splenic function has been reported before [15]. Importantly, in a small subset of patients, we found higher HVPG values in patients with FH as compared to patients without FH, but the results were not statistically significant most likely to the small number of patients with available HVPG measurements.

Our study has certain limitations, most of them are inherent to real-world observational studies: we only studied hematological parameters (pitted erythrocytes, Howell-Jolly bodies), but did not use scintigraphic methods like the splenic uptake of $^{99m}$Tc-labelled, heat-altered erythrocytes or further functional immunological tests to determine the spleen function. However, the sensitivity and specificity of pitted erythrocytes were shown to have a high correlation with scintigraphic methods for the determination of spleen function in previous studies [17]. Furthermore, pitted erythrocytes are a marker of the phagocytic spleen function, but to study the immunological spleen function, vaccination responses and B-cell subsets should be

measured [18]. Second, it should be noted that other authors reported much lower pitted erythrocyte counts in children with sickle cell anemia who had an absent spleen [99mTc]-uptake in scintigraphy [15] and the identification of pitted erythrocytes may be subject to an inter-observer bias: Thus, we cannot rule that our study overestimated the percentage of pitted erythrocytes. Third, follow-up data of the patients were collected by simple chart review only, as standardized follow-up visits were not performed. Also, not all of the infections of the study participants might have been recorded during the follow-up period. Most likely due to study design and lack of statistical power, we did not find an association of functional hyposplenism with infection rates, all-cause mortality, or infection-related mortality.

In conclusion, our study demonstrates that functional hyposplenism is frequently detectable in cirrhotic patients independently from the etiology of cirrhosis. Further research is needed to study the impact of functional hyposplenism on infection rates and most importantly on the clinical course of patients with liver cirrhosis. Vaccination guidelines for patients with progressed liver disease might have to be updated in accordance with the recommendations for asplenic patients.

## Author Contributions

**Conceptualization:** Malte H. Wehmeyer, Julian Schulze zur Wiesch.

**Data curation:** Malte H. Wehmeyer, Harsha Sekhri.

**Formal analysis:** Malte H. Wehmeyer, Harsha Sekhri.

**Investigation:** Malte H. Wehmeyer, Harsha Sekhri, Raluca Wroblewski, Antonio Galante.

**Methodology:** Malte H. Wehmeyer, Harsha Sekhri, Raluca Wroblewski, Julian Schulze zur Wiesch.

**Writing – original draft:** Malte H. Wehmeyer, Harsha Sekhri.

**Writing – review & editing:** Raluca Wroblewski, Antonio Galante, Thomas Meyer, Ansgar W. Lohse, Julian Schulze zur Wiesch.

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
