## [Decision Letter · Decision Letter 0]

15 Mar 2022

PONE-D-22-03651Frequent detection of functional hyposplenism via assessment of pitted erythrocytes in patients with advanced liver cirrhosisPLOS ONE

Dear Dr. Wehmeyer,

Thank you for submitting your manuscript to PLOS ONE. After careful consideration, we feel that it has merit but does not fully meet PLOS ONE’s publication criteria as it currently stands. Therefore, we invite you to submit a revised version of the manuscript that addresses the points raised during the review process.

We look forward to receiving your revised manuscript.

Kind regards,

Gopal Krishna Dhali, MBBS, MD, DM

Academic Editor

PLOS ONE

Journal Requirements:

Reviewers' comments:

Reviewer's Responses to Questions

**Comments to the Author**

1. Is the manuscript technically sound, and do the data support the conclusions?

Reviewer #1: Yes

Reviewer #2: Yes

2. Has the statistical analysis been performed appropriately and rigorously? 

Reviewer #1: Yes

Reviewer #2: Yes

3. Have the authors made all data underlying the findings in their manuscript fully available?

Reviewer #1: No

Reviewer #2: Yes

4. Is the manuscript presented in an intelligible fashion and written in standard English?

Reviewer #1: Yes

Reviewer #2: Yes

5. Review Comments to the Author

Reviewer #1: This is a good paper overall and the authors should be commended for their hard work. Few comments / suggestions:

1. Figure 2 is not adding too much value or information - suggest removing it / adjusting the table.

.

2. Further details on interference contrast microscopy should be added to the methods section - what microscope etc.

.

3. You should add some pictures (?Appendix) of the pitted erythrocytes - the threshold of defining functional hyposplenism from this group is 15% and they have justified that - this is fine. Some other groups, including ours, have used a threshold of 4% and seeing so many patients with a pitted erythrocyte count >4% makes me wonder about over-counting some cells as pitted erythrocytes when they were not? [see BABY-HUG trial sickle cell, 10.1182/blood-2010-04-278747]. Hence, having pictures of pitted erythrocytes would help. Some of the referenes the authors used for arguing their threshold of 15% date back to 1976 - slightly outdated [reference 19].

.

4. Agree with conclusions. The discussion is interesting. Do the authors have any other mechanistic reasons for hyposplenism in liver disease to comment on?

Reviewer #2: The manuscript by Wehmeyer et al focuses on hyposplenism in advanced cirrhosis patients. They used measurement of pitted RBCs as a measure of splenic function. A few clarification or further details on some issues could be helpful, along with minor corrections as mentioned below:

Abstract

Use full forms of abbreviations at first use

Structure the abstract into subheadings

Association with portal hypertension- not clear (from abstract)

Introduction

Mention one of the studies on PE in detail, and as a function of FH.

PE should be mentioned as one of the indirect markers of FH.

Methods

Specify following

Investigations done for etiologic work up of cirrhosis.

Other causes of PE ruled out?

Line 90- („pitted erythrocytes“) were… - correct the inverted comma

Line 95-‘percentage of erythrocytes with these abnormalities was calculated’- after counting _____ RBC.

Vaccination/anti-microbial policy in the patients.

It will be very helpful, if data is provided on past and study period infections in the study subjects- a better correlation of the overall outcome.

Results- well described

Mention the following in results/tables

Outcomes of the patients- with respect to infection rate observed in past and study period, infection related morbidity & mortality, other causes of mortality etc

Correlation with portal hypertension

Discussion- well written

Limitation of PE as a measure of splenic function.

Figures

Mention the graph/ statistical test depicted to show the results

6. PLOS authors have the option to publish the peer review history of their article (what does this mean?). If published, this will include your full peer review and any attached files.

Reviewer #1: **Yes: **Abhinav Mathur

Reviewer #2: No

---

## [Author Response · Author response to Decision Letter 0]

16 Jun 2022

Reviewer #1: This is a good paper overall and the authors should be commended for their hard work. Few comments / suggestions:

- We thank reviewer #1 for their benevolent reception, critical review, and acknowledgment. Please find a detailed response to each suggestion below:

1. Figure 2 is not adding too much value or information - suggest removing it / adjusting the table.

- We agree with reviewer #1, therefore we removed (former) Figure 2 in the revised manuscript as the data are already presented in the results section and Table 1.

2. Further details on interference contrast microscopy should be added to the methods section - what microscope etc.

- We thank reviewer #1 for this suggestion. We added the respective information in the methods section.

3. You should add some pictures (?Appendix) of the pitted erythrocytes - the threshold of defining func-tional hyposplenism from this group is 15% and they have justified that - this is fine. Some other groups, including ours, have used a threshold of 4% and seeing so many patients with a pitted erythrocyte count >4% makes me wonder about over-counting some cells as pitted erythrocytes when they were not? [see BABY-HUG trial sickle cell, 10.1182/blood-2010-04-278747]. Hence, having pictures of pitted erythrocytes would help. Some of the referenes the authors used for arguing their threshold of 15% date back to 1976 - slightly outdated [reference 19].

- Thank you for the suggestion. We added an image of pitted erythrocytes (and HJB) as Figure 1 in the revised manuscript. Furthermore, we included the referenced paper in our introduction sec-tion. We now discuss a potential observer-based bias in the revised manuscript.

4. Agree with conclusions. The discussion is interesting. Do the authors have any other mechanistic rea-sons for hyposplenism in liver disease to comment on?

- Thank you. Previously, FH has been attributed to alcohol abuse in cirrhotic patients, which has al-ready been discussed in the original manuscript. However, our data indicate that patients with other etiologies of cirrhosis also suffer from FH. Our results do not implicate another mechanistic reason for hyposplenism other than chronic portal hypertension. There might be additional func-tional immunological reasons (reduced clearance of microbial translocation etc.), however, we feel that such a discussion would be too hypothetical and not based on our data. Therefore, we would rather not include these valid discussion points in the discussion of the current manuscript. 

Reviewer #2: The manuscript by Wehmeyer et al focuses on hyposplenism in advanced cirrhosis patients. They used measurement of pitted RBCs as a measure of splenic function. A few clarification or further details on some issues could be helpful, along with minor corrections as mentioned below:

- We would also tank reviewer #2 for their helpful and fair review f the current study.

Abstract

Use full forms of abbreviations at first use.

- We modified the abstract according to the reviewer´s suggestions. However, to improve the read-ability of the abstract, “functional hyposplenism” and “pitted erythrocytes” were spelled out throughout the abstract of the revised manuscript.

Structure the abstract into subheadings.

- We structured the manuscript with the subheadings “background”, “methods”, “results” and “conclusions”.

Association with portal hypertension not clear (from abstract).

- We agree with reviewer #2. The association of portal hypertension with liver stiffness measure-ments (by transient elastography) is now mentioned in the conclusion section of the revised ab-stract.

Introduction

Mention one of the studies on PE in detail, and as a function of FH.

- We thank the reviewer for this suggestion. We provide these details in the revised manuscript (Lammers et al. Am J Hematol 2012). 

PE should be mentioned as one of the indirect markers of FH.

- We agree with reviewer #2. Therefore, we changed the sentence “Therefore, the occurrence of pitted erythrocytes is associated with poor splenic function”, to “…, pitted erythrocytes are re-garded a reliable indirect marker of functional hyposplenism”.

Methods

Specify following:

Investigations done for etiologic work up of cirrhosis.

- We added the appropriate information in the revised manuscript.

Other causes of PE ruled out?

- Patients with a history of hematological co-morbidities and splenectomized patients were not in-cluded in the cirrhosis group. We added this clarification to the methods section of the revised manuscript. However, as portal vein thrombosis (PVT) is considered an alternative cause of func-tional hyposplenism and patients with liver cirrhosis often suffer from PVT, we further studied a possible association between FH and PVT: 4/28 patients with FH and 12/89 patients without FH had a PVT (P=1). We included this finding in table 1 of the revised manuscript.

Line 90- („pitted erythrocytes“) were… - correct the inverted comma.

- We corrected the inverted comma in the revised manuscript.

Line 95-‘percentage of erythrocytes with these abnormalities was calculated’- after counting _____ RBC.

- We added clarification of the calculation of the PE percentage.

Vaccination/anti-microbial policy in the patients.

- We evaluated the vaccination rates against pneumococci in our cohort. The corresponding para-graphs of the methods and results section of the manuscript were revised accordingly. Further-more, the revised discussion section now clearly emphasizes the low vaccination rates. As this was an observational study, we did not change the vaccination schedule for patients in our study (in fact, this would be a clinical trial as per German jurisdiction). 

It will be very helpful, if data is provided on past and study period infections in the study subjects- a better correlation of the overall outcome.

- We thank reviewer #2 for this important suggestion. Please see our response to the corresponding suggestion for the results section for further details.

Results:

Well described. Mention the following in results/tables:

Outcomes of the patients- with respect to infection rate observed in past and study period, infection relat-ed morbidity & mortality, other causes of mortality etc.

- We agree with reviewer #2. We collected the follow-up data by chart review of all patients. Inter-estingly, we were able to collect follow-up data from nearly all patients (112/117). However, due to the small sample size and due to study design (observational study without a pre-planned fol-low-up period or scheduled patient visits), we did not find a statistically significant association be-tween FH and infection rates or other related variables. These limitations are now discussed transparently in the revised discussion section as well. Furthermore, the methods section was up-dated with regard to the follow-up data.

Correlation with portal hypertension.

- We re-organized the results section to further emphasize the evaluation of PE counts with portal hypertension. Transient elastography measurements were the only indicator of portal hyperten-sion which was associated with the presence of functional hyposplenism, while the presence of ascites or esophageal varices was not associated with functional hyposplenism. Furthermore, we found an association between PE counts and the presence of portal hypertension according to the most recent Baveno VII criteria. In 13 patients hepatic venous pressure gradients were available from the time of study inclusion and patients with FH had higher HVPG. Due to the low number of patients (n=13), the results were statistically not significant. The discussion section was also up-dated according to these results.

Discussion:

Well written. Limitation of PE as a measure of splenic function.

- Thank you. The revised discussion section of the manuscript now clarifies, that PE is regarded as a marker of the phagocytic spleen function and that additional studies of immunization responses and B cell subsets are needed to evaluate the immunological function of the spleen.

Figures:

Mention the graph / statistical test depicted to show the results.

- We added the respective information to the figure legend.

---

## [Decision Letter · Decision Letter 1]

4 Jul 2022

Frequent detection of functional hyposplenism via assessment of pitted erythrocytes in patients with advanced liver cirrhosis

PONE-D-22-03651R1

Dear Dr. Wehmeyer,

We’re pleased to inform you that your manuscript has been judged scientifically suitable for publication and will be formally accepted for publication once it meets all outstanding technical requirements.

Kind regards,

Gopal Krishna Dhali, MBBS, MD, DM

Academic Editor

PLOS ONE

**Comments to the Author**

1. If the authors have adequately addressed your comments raised in a previous round of review and you feel that this manuscript is now acceptable for publication, you may indicate that here to bypass the “Comments to the Author” section, enter your conflict of interest statement in the “Confidential to Editor” section, and submit your "Accept" recommendation.

Reviewer #2: All comments have been addressed

2. Is the manuscript technically sound, and do the data support the conclusions?

Reviewer #2: Yes

3. Has the statistical analysis been performed appropriately and rigorously? 

Reviewer #2: Yes

4. Have the authors made all data underlying the findings in their manuscript fully available?

Reviewer #2: No

5. Is the manuscript presented in an intelligible fashion and written in standard English?

Reviewer #2: Yes

6. Review Comments to the Author

Reviewer #2: The authors have made all the necessary modifications as desired by the reviewers. I would like to congratulate them for this improved version of the manuscript.

7. PLOS authors have the option to publish the peer review history of their article (what does this mean?). If published, this will include your full peer review and any attached files.

Reviewer #2: No

---

## [Editor Report · Acceptance letter]

7 Jul 2022

PONE-D-22-03651R1 

Frequent detection of functional hyposplenism via assessment of pitted erythrocytes in patients with advanced liver cirrhosis 

Dear Dr. Wehmeyer:

I'm pleased to inform you that your manuscript has been deemed suitable for publication in PLOS ONE. Congratulations! Your manuscript is now with our production department. 

Kind regards, 

on behalf of

Dr. Gopal Krishna Dhali 

Academic Editor

PLOS ONE